# Technical determinants of success in professional women's soccer: A wider range of variables reveals new insights

**Laura M. S. de Jong**[1]*, **Paul B. Gastin**[2], **Maia Angelova**[3], **Lyndell Bruce**[1], **Dan B. Dwyer**[1]

**1** Centre for Sport Research, Deakin University, Geelong, Australia, **2** Sport and Exercise Science, La Trobe University, Melbourne, Australia, **3** Data to Intelligence Research Centre, School of Information Technology, Deakin University, Geelong, Australia

* l.dejong@deakin.edu.au

**Data Availability Statement:** The data that was used for this study was acquired from a third-party, formerly Opta Sports, now Stats Perform. The data was provided under a license agreement with Opta

## Abstract

Knowledge of optimal technical performance is used to determine match strategy and the design of training programs. Previous studies in men's soccer have identified certain technical characteristics that are related to success. These studies however, have relative limited sample sizes or limited ranges of performance indicators, which may have limited the analytical approaches that were used. Research in women's soccer and our understanding of optimal technical performance, is even more limited (n = 3). Therefore, the aim of this study was to identify technical determinants of match outcome in the women's game and to compare analytical approaches using a large sample size (n = 1390 team performances) and range of variables (n = 450). Three different analytical approaches (i.e. combinations of technical performance variables) were used, a data-driven approach, a rational approach and an approach based on the literature in men's soccer. Match outcome was modelled using variables from each analytical approach, using generalised linear modelling and decision trees. It was found that the rational and data-driven approaches outperformed the literature-driven approach in predicting match outcome. The strongest determinants of match outcome were; scoring first, intentional assists relative to the opponent, the percentage of shots on goal saved by the goalkeeper relative to the opponent, shots on goal relative to the opponent and the percentage of duels that are successful. Moreover the rational and data-driven approach achieved higher prediction accuracies than comparable studies about men's soccer.

## Introduction

Professional women's soccer is a relatively young sport compared to its men's counterpart, with the first official World Cup held in 1991 [1]. Currently, it is experiencing rapid growth, both in numbers of participants [2–4] and professionalism [5]. Likewise, research attention is growing; however, this has mainly focused on the physical aspects of the game [6–8] and only a limited number of studies have investigated the technical side [9–11]. Understanding optimal

Sports/Stats Perform, and the data is also subject to an approved research ethics application from our University. The terms of the license agreement prevents us from sharing the raw data we used for this analysis. Our ethical approval also prevents us from sharing any data in any way that could be re-identified. The metadata and the (score) data itself would allow someone else to re-identify teams and possibly players. However, with the information below access to the data should be possible from the third-party. The data acquired were so called 'excel dumps' of team level statistics per match of the following leagues and tournaments: the American National Women's Soccer League (NWSL) (2016-2018 seasons); the British Football Association Women's Super League (FAWSL) (2015/16-2017/18 seasons); the 2013 and 2017 UEFA Women's Euros Championships; the 2011 and 2015 FIFA Women's World Cups. Access to the data can be organised by contacting Stats Perform https://www.statsperform.com/contact/. The authors had no special access privileges.

**Funding:** The author(s) received no specific funding for this work.

**Competing interests:** The authors have declared that no competing interests exist.

technical performance enhances training programming and match strategy planning [12]. Of those that have examined technical performance, it has been shown that a quarter of goals are made from a cross [9], scoring first increases the likelihood of winning [10] and that free kicks within 7 m of the penalty circle should be taken as a direct free kick (shot on goal) to increase goal scoring [11]. Another study included some technical variables and found that duels both in offense and defence, pass accuracy and effective execution of dead-ball moments were significant contributors to victory [13]. These few studies give us the first insights into technical actions required to increase the likelihood of scoring or winning in women's soccer. In men's soccer, several studies have investigated technical performance related to success. It has been found that technical actions such as shots on goal [14–17], passes [18, 19] and assists [15, 17] were related to winning and these variables could also be important contributors to success in women's soccer.

The identification of optimal technical performance in women's soccer should use the best available analytical methods. This can be informed by an exploration of the strengths and limitations of analytical methods that have been used previously in soccer. Some previous studies may have been limited in sample size, as they were based on only one season or one tournament [9–11, 13, 14, 16, 17, 19–21], or a few seasons of a single tournament or league [15, 22]. While these sample sizes may not violate the statistical assumptions of the analytical methods that were used, they may limit the generalisability of their conclusions.

Some studies may also have been limited in the range of technical variables used in their analyses. The number of technical variables used in the limited studies in women's soccer was relatively small (n = 1–9), even when combining the studies together [9–11]. Most previous studies in men's soccer used 15–25 technical variables [14–17, 19–22] in their analysis and all except one [17] did not provide a clear rationale for the use of those particular variables. Previous research in Australian Rules football has shown that utilising a wide variety of variables gives more accurate match outcome models [23] and this could apply to modelling of match outcome in soccer as well. For some studies, the limited number of technical variables used may be ascribed to the number of variables that were available. For other studies, the number of variables used may also be a result of decisions based on the assumptions of statistical models chosen for analysis. The classification accuracy or predictive accuracy of models increases when the number of variables increases but so does the likelihood of overfitting [24]. Other than the variety of variables, it was found in Australian Rules football that variables in their relative form (i.e. difference in value to the opposing team in the same game) are better predictors than absolute values [23].

A further limitation of some studies of technical performance relates to the statistical models used. Some studies use statistical models that assume independence of observations [11, 13, 14, 20], which may not be met when observations from all teams appear repeatedly within a dataset. Moreover, to the best of our knowledge, none of the studies in the literature validated the performance of their modelling on a new data set or a part of the data set that is kept apart from the model development phase. This is a common practice in data mining and has also been suggested as an appropriate method to reduce overfitting [24]. Examples are splitting the dataset into a model training and test-set or using a 10-fold cross-validation.

Little is known about optimal technical performance in women's soccer and the findings of research in men's soccer probably has restricted transferability to women's soccer. It is known that there is a difference in physical performance between genders [25, 26] which could also be the case for technical performance. The limitations of research regarding technical performance in soccer, taken with the limited transferability of findings from the men's game to women's game, indicate the scope for further research. Given the apparent increase in the professionalism of women's soccer, there is a need to improve our knowledge of optimal technical performance and how it may differ from men's soccer. Therefore, our aim was two-fold:

understand the technical determinants of success in women's soccer using a large sample size and a large number of variables, and identify analytical approaches that provide the most accurate and reliable information based on the prediction accuracy of the models.

## Materials and methods

### Data collection

Data was acquired from Opta Sports (London, United Kingdom) and covered the British Football Association Women's Super League (FAWSL) 2015, 2016 and 2017/18 seasons, the American National Women's Soccer League (NWSL) 2016–2018 seasons, the 2013 and 2017 Union of European Football Associations (UEFA) Women's Euros Championships (EC) and the 2011 and 2015 Fédération Internationale de Football Association (FIFA) Women's World Cups (WC). This resulted in 695 matches and 1390 samples (sets of technical variable values per team per match) (see Table 1). Data consisted of team aggregates per match of all events that Opta Sports collects (i.e. 266 variables per team in each game). A detailed description of the events can be found on the Opta Sports website [27]. Empty variables were removed (n = 3), and match outcome (Win/Loss/Draw) was calculated based on the goals scored per team relative to the opponent. All analyses were done in R [28] and ethical exemption was received from the Deakin University Human Research Ethics Committee (2018–392) because of the use of pre-existing non-identifiable data.

### Analysis

**Outlier detection.** A search for outlying samples was done using a K-nearest neighbour outlier detection method [29] by adjustment of the do_*knno()*-function of the *adamethods*-package [30]. This identified 14 samples (7 matches) standing out from the rest of the data. Further investigation identified that all those matches were valid data points and it was decided to retain them in the dataset. Numerical variables were checked for normality, which was violated. An analysis for typographical errors within variables was done with the Adjusted Outlyingness, a method suitable for skewed data [31], using the *adjbox()*-function of the *robustbase*-package [32]. No errors were identified. In order to identify the relationship between technical performance and match outcome, 31 variables that were a function of score were removed, examples are: winning goal, goals from corners and goals conceded. All remaining variables that were not considered sample identifiers were expressed in absolute and relative forms (value relative to the opposing team in each match).

**Table 1. Distribution of matches per league or tournament.**

| League/Tournament | Team performances |
| --- | --- |
| EC 2013 | 50 |
| EC 2017 | 62 |
| FAWSL 2015 | 112 |
| FAWSL 2016 | 144 |
| FAWSL 2017–18 | 180 |
| NWSL 2016 | 206 |
| NWSL 2017 | 246 |
| NWSL 2018 | 222 |
| WC 2011 | 64 |
| WC 2015 | 104 |
| **Total** | **1390** |

**Outline of analysis process.** Initially, a feature selection process was executed, followed by two phases of modelling and then a comparison of the results for the different models. During the modelling in each phase, three different approaches (i.e. sets of technical variables) were used to identify the relationship between technical performance and match outcome. This was to allow comparability with the previous literature, but also to determine whether newer approaches may provide better outcomes. The first approach was data-driven (DDA) and for the first phase, it was decided that all 450 variables were used for feature selection. For the second phase, 152 variables were included. The second approach was a rational approach (RA) in which two authors selected a broad range of variables (n = 156 in phase one and n = 74 in phase two) that were considered relevant to coaches. This was done to reduce overfitting and increase the practical application. In the third approach, variables were selected that had been used previously in the literature on men's soccer (n = 43 in phase one and n = 16 in phase two) [14–17, 22] in order to make some form of comparability possible. Henceforth, this will be called the literature-driven approach (LDA). After a multicollinearity check and the first phase of modelling, a second phase of modelling was executed. For this second phase, all variables that were considered pseudo-score related were removed. Pseudo-score related in this context meant variables that will almost certainly precede a goal scored or by their definition mean that a goal was scored, such as 'shots on goal' and 'scoring the first goal'. This step was intended to allow us to determine whether these variables are/not important in match outcome and then exclude them from further modelling. This allowed the identification of aspects of technical performance that might not have been identified if pseudo-score related were retained. This resulted in 152 variables being used for feature selection in the DDA, 74 in the RA and 16 in the LDA. A flowchart of the full analysis process can be found in Fig 1.

**Feature selection.** For all three approaches (DDA, RA and LDA), variables were checked for multicollinearity and selected/excluded using correlation-based feature selection [33] applying the *findCorrelation*-function [34] and a cut-off Pearson correlation coefficient of >0.90. For phase one, this left 367 variables in the DDA, 140 in the RA and 41 in the LDA and for phase two, 52 variables in the DDA, 74 in the RA and 16 in the LDA for further analysis. Thereafter, drawn matches (n = 292) were removed from further analysis since they can neither be considered clearly successful nor unsuccessful.

Following a wrapper method, recursive feature selection (RFS) using a random forest [35], was applied using the *rfe*-function and a 10-fold cross-validation (CV) repeated 5 times in the *caret*-package [34] to reduce the number of variables for the DDA and RA in both phases. This was done to avoid reduced model accuracy caused by a high number of variables in the modelling phase (i.e. too many variables risks adding noise to the model and or overfitting). In the first phase for both approaches, the RFS showed that the optimal number of variables was around 50 and therefore it was decided for each approach, to select the 50 variables with the highest score on the variable importance measure. The same analysis was done in the second phase, but this gave an optimal number of 30 for the DDA and 60 variables for the RA. Following this, the same procedure was applied to determine a variable importance measure for all variables in the LDA for both phases. The first 20 variables after feature selection for the three different approaches can be found in Table 2.

**Modelling.** For the modelling phase, the samples were split into a training and testing set, with an 80:20 ratio. For each approach a binominal generalized linear model (GLM) with lasso penalized maximum likelihood [36] using the *glmnet()*-function of the *caret*-package [34] was utilised. In order to compare a partially interpretable model with an interpretable model, a decision tree (DT) was applied for each approach using the *rpart()*-function of the *rpart*-package [37] within *caret*. All modelling was done with a 10-fold CV repeated 10 times. Model

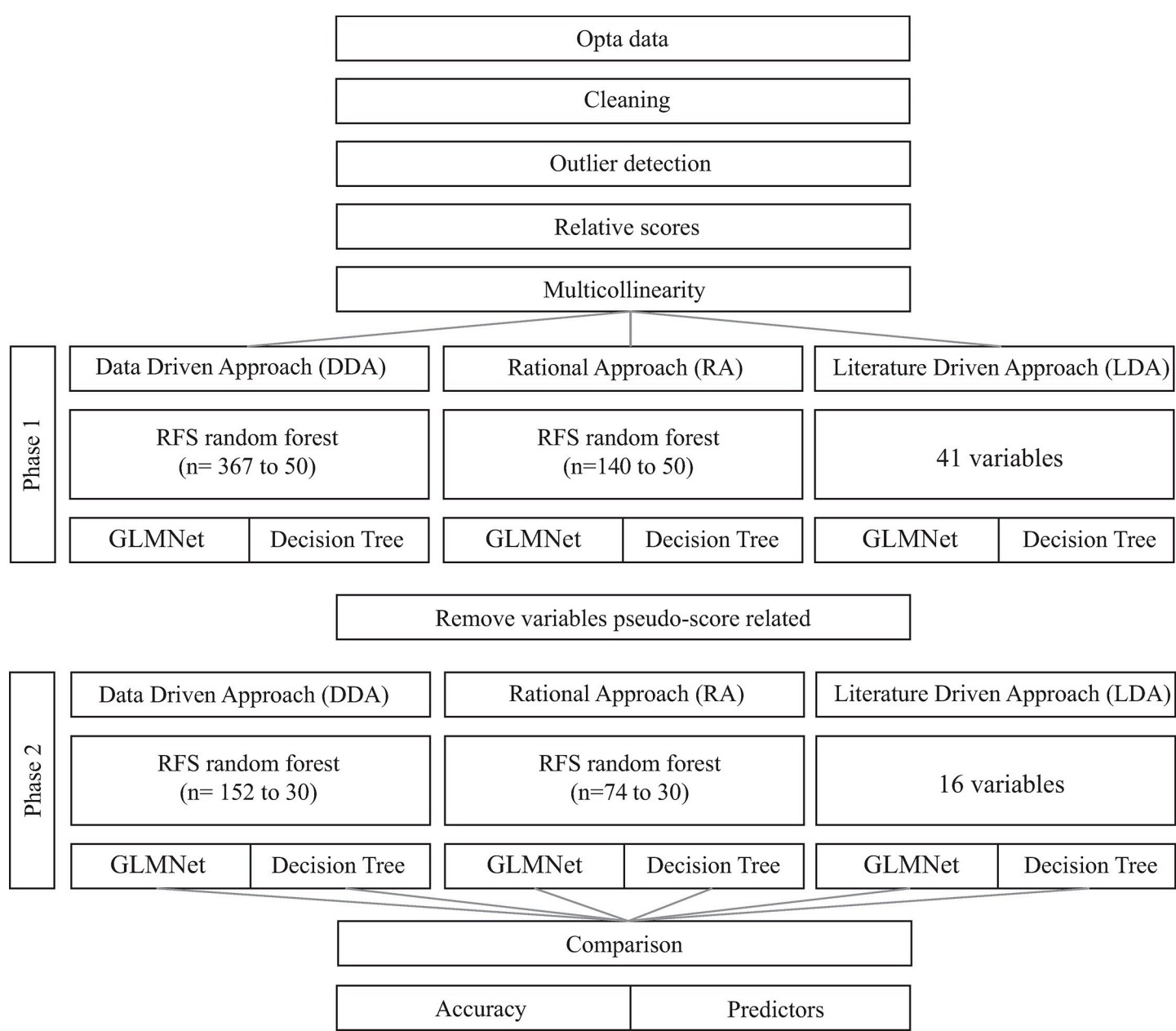

**Fig 1. Flowchart of the analysis process.** Details of the techniques are explained in the text.

performance was evaluated by determining their prediction accuracy using the testing data. Accuracies are presented on scale 0–1 with a 95% Confidence Interval (CI).

## Results

The dataset provided by the RA in the first phase (i.e. inclusive of pseudo-score related variables) allowed the creation of the most accurate models from both the GLMs (0.99 CI 0.97–1.00) and the DTs (0.89 CI 0.84–0.92) (see Table 3). Moreover, of all 12 models, the GLM using the RA in the first phase is the most accurate. For all approaches, the models that retained pseudo-score related variables, outperformed models without. In both phases, the

**Table 2. The most valuable 20 technical variables identified in feature selection for the three analytical approaches.**

| Rank | Data-Driven Approach | Rational Approach | Literature-Driven Approach |
|------|----------------------|-------------------|----------------------------|
| 1 | First Goal | Relative Percentage Shots On Saved | Relative Shots On |
| 2 | Relative Intentional Assist | Relative Intentional Assist | Shots On |
| 3 | Assists | First Goal | Relative Total Shots |
| 4 | Relative Big Chances On Target | Percentage Shots On Saved | Relative Aerial Duels Won |
| 5 | Relative Shots On Conceded Inside Box | Relative Shots On From Inside Box | Total Shots |
| 6 | Intentional Assist | Intentional Assist | Relative Successful Crosses Corners |
| 7 | Relative Attempts Open Play On Target | Shots On | Total Clearances |
| 8 | Relative Big Chance Created | Relative Total Shots | Relative Total Passes |
| 9 | Relative Right Foot Shots On Target | Shots On From Inside Box | Relative Total Clearances |
| 10 | Shots On Conceded | Relative Saves Made | Relative Offsides |
| 11 | Shots On Conceded Inside Box | Relative Recoveries | Relative Corners Conceded |
| 12 | Big Chances Faced | Total Shots | Relative Unsuccessful Crosses Corners |
| 13 | Big Chances On Target | Relative Shots On Target Outside Box | Successful Crosses Corners |
| 14 | Shots On From Inside Box | Relative Second Assists | Corners Conceded |
| 15 | Relative Touches Open Play Opponent Box | Percentage Successful Duels | League Or Tournament |
| 16 | Relative Shooting Accuracy Right Foot | Relative Successful Dribbles | Total Passes |
| 17 | Relative Recoveries | Saves Made | Unsuccessful Crosses Corners |
| 18 | Relative Defensive Aerial Duels Won | Relative Percentage Total Successful Passes | Corners Taken |
| 19 | Total Shots Conceded | Relative Successful Crosses Corners | Aerial Duels Won |
| 20 | Relative Second Assists | Relative Aerial Duels Lost | Offsides |

All variables were used in the first phase of analysis, which included variables that have a pseudo-relationship with scoring. The pseudo-score related variables (underlined) were removed in the second phase of the analysis.

dataset provided by the LDA gives the lowest prediction accuracy, both for the GLM (phase 1: 0.78 CI 0.72–0.84; phase 2: 0.65 CI 0.58–0.71) and the DT (phase 1: 0.75 CI 0.68–0.80; phase 2 0.66 CI 0.59–0.72). The accuracies of the models based upon data from the LDA in the second phase, perform just slightly higher than a 50% chance of win/loss.

In the second analysis phase (i.e. where pseudo-score related variables were excluded), the DDA allowed the creation of a more accurate model GLM (0.87 CI 0.82–0.91), compared to both the RA (0.78 CI 0.72–0.83) and the LDA (0.65 CI 0.58–0.71) (Table 3). In general, the DTs did not perform as well as the GLMs, although there were combinations of approach and phase where they performed almost equally.

When looking at the specific variables that were included in the models in the first analysis phase, the most important technical action for winning a match in women's soccer is scoring the first goal. With 20.3% of the matches won with 1–0 (141 out of 695 matches) and 55.3% of teams who scored a goal also winning their game (992 teams scored one or more goal, and 549 of those teams won their game). Following this, the next most important technical actions are intentional assists relative to the opponent and percentage of shots on goal saved by the goalkeeper relative to the opponent. The different models relied on different combinations of technical actions (e.g. relative big chances on target, relative right foot shots on target, relative shots on conceded inside box, relative percentage shots on goal saved and relative shots on from inside box) to explain match outcome. The LDA only had one significant contributor, namely shots on goal relative to the opponent, both in the GLM and the DT. For all models, variables that are expressed relative to their opponent are stronger contributors to match outcome than variables in their absolute form.

**Table 3. The structure and performance of the generalised linear (GLM) and Decision Tree (DT) models in both phases of the analysis.**

| Phase 1 | Data Driven Approach (DDA) | | Rational Approach (RA) | | Literature Driven Approach (LDA) | |
|---|---|---|---|---|---|---|
| | Variable | Coefficient | Variable | Coefficient | Variable | Coefficient |
| GLM Predictors | First Goal | -1.47 | First Goal | -1.24 | Relative Shots On Goal | -0.20 |
| | Relative Intentional Assist | -0.58 | Relative Percentage Shots On Goal Saved | -0.95 | | |
| | Relative Big Chances On Target | -0.07 | Relative Intentional Assist | -0.49 | | |
| | Relative Right Foot Shots on Target | -0.01 | Relative Shots On From Inside Box | -0.06 | | |
| | Relative Shots On Conceded Inside Box | 0.01 | | | | |
| GLM Accuracy (95% CI) | 0.96 (0.93, 0.98) | | 0.99 (0.97, 1.00) | | 0.78 (0.72, 0.84) | |
| DT Accuracy (95% CI) | 0.88 (0.83, 0.92) | | 0.89 (0.84, 0.92) | | 0.75 (0.68, 0.80) | |
| **Phase 2** | | | | | | |
| GLM Predictors | Relative Intentional Assist | 0.75 | Percentage Successful Duels | -2.01 | Relative Aerial Duels Won | -0.05 |
| | Assists | 0.17 | Relative Intentional Assist | -0.69 | Relative Offsides | -0.04 |
| | Relative Touches Open Play Opponent Box | -0.002 | Intentional Assist | -0.24 | Relative Total Passes | -0.001 |
| | | | Relative Successful Dribbles | -0.02 | | |
| | | | Relative Recoveries | -0.01 | | |
| | | | Relative Aerial Duels Lost | 0.003 | | |
| GLM Accuracy (95% CI) | 0.87 (0.82, 0.91) | | 0.78 (0.72, 0.83) | | 0.65 (0.58, 0.71) | |
| DT Accuracy (95% CI) | 0.76 (0.70, 0.82) | | 0.77 (0.70, 0.82) | | 0.66 (0.59, 0.72) | |

Phase 1 of the analysis included pseudo-score related variables, whereas in phase 2 they were excluded. There were also three analytical approaches that reflect different sets of variables that were used for modelling. Accuracies presented on scale 0–1 with 95% Confidence Interval (CI).

For the second phase of the analysis, in both the DDA and the RA, relative intentional assists are still strong contributors to the model. They are the first contributor in the data-driven approach GLM and DT, rational approach DT and the second in the rational approach GLM. Other important contributors to success in this second phase are intentional assists in their absolute form, normal assists and the percentage of successful duels. The coefficients of the literature-driven approach GLM are relatively small. Finally, for both phases in the LDA, intentional assists do not appear in the models since this variable has not been identified in the existing literature in men's soccer and therefore was not included in the LDA. The DT for the DDA of the second phase is shown in Fig 2, and it can be seen that a value increment of one can make a difference.

## Discussion

This study aimed to identify technical determinants of match outcome in women's soccer and to compare different analytical approaches. The main findings are that scoring first and having more assists than an opponent are very important in women's soccer. When pseudo-score related variables are removed, technical actions such as successful duels, become important, which have not previously been identified in men's soccer. In addition, this study highlights the value of using a larger number of games and variables in this type of analysis, as they tend to produce improvements in model accuracy and reveal novel aspects of technical performance.

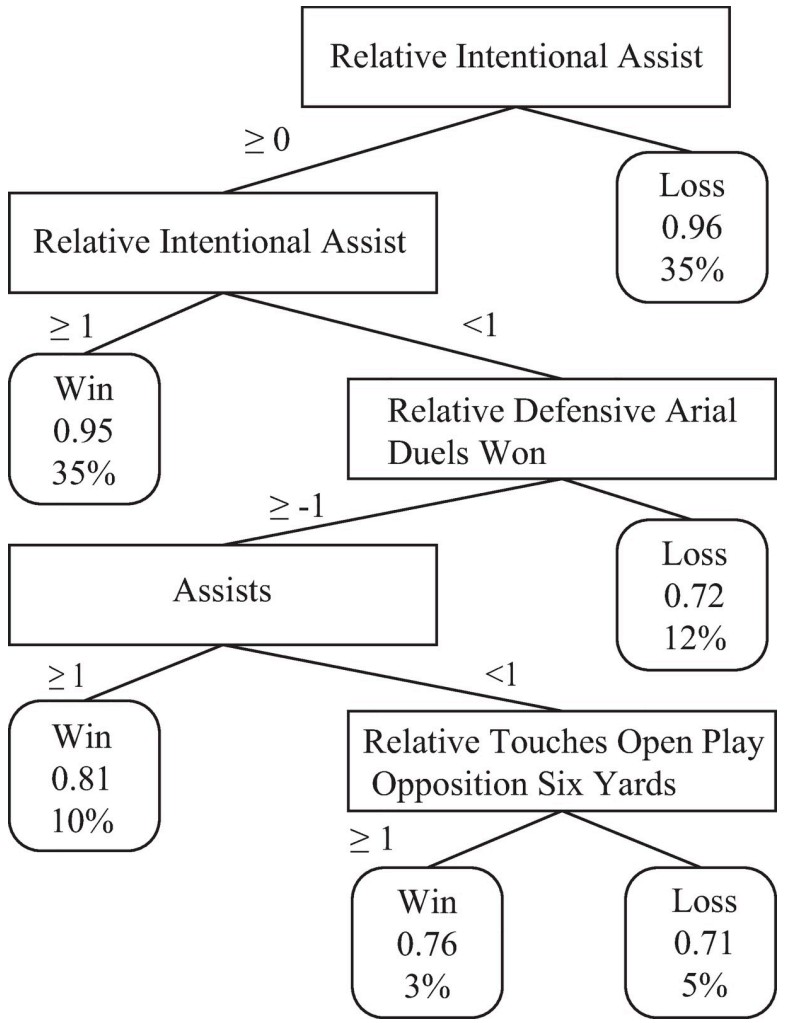

**Fig 2. Decision tree for the data-driven approach in the second phase.** The tree indicates the association between the values of the key performance indicators and the likely match outcome. This model predicted match outcome with a classification accuracy of 76%. Percentages mentioned are the proportion of the total dataset. Probability shown is the likelihood of a match with those conditions falling into the final category. The number on the branches of the tree indicates the cut off value for the technical action.

The most important contributor to success in women's soccer is scoring the first goal. It is not clear why this occurs but it may relate to the psychological impact on both teams. This aligns with the findings of previous research in women's [10] and men's soccer [38, 39]. Even though this might be hard to implement in training and is very dependent on the match situation, awareness of this fact is important to coaches. It could be included in a women-specific match strategy or incorporated in the tactical plan for after the first goal. Another important contributor to success in women's soccer is intentional assists relative to the opponent. Intentional assist refers to a player making an intentional pass to a teammate who then makes a shot on goal, so there is no deflection of the pass before the shot [40]. This finding is novel and might be women-specific, as our models suggest that assists were more important than shots on goal. However, given that shots on goal precedes assists, it is difficult to definitively separate the importance of these two actions. Shots on goal did not appear in 11 of our 12 models whereas assists or intentional assists were a strong contributor in 8 of our 12 models. The

number of goals saved by the goalkeeper relative to the opponent was also an important determinant of match outcome. Furthermore, to our knowledge, no previous study has presented evidence that the performance of the goalkeeper is an important determinant of match outcome.

The second phase of modelling excluded pseudo-score related variables that were considered to be difficult to train in a game-play setting. The analysis revealed that relative intentional assists are still very important. In addition, intentional assists, assists, percentage successful duels, relative aerial duels won, relative defensive aerial duels won, relative total passes and relative offsides were important contributors to match outcome. Interestingly most of those variables could be considered typical of an aggressive style of offensive and defensive play. These findings could imply that a more aggressive style of play is associated with winning matches in women's soccer, which could be incorporated in the women-specific match strategy.

The literature-driven approach in the present study was designed to permit the comparison of men's and women's soccer. Previous studies in men's soccer showed that passes [16–18], shots on goal [14–19, 41], and assists [15, 17] were important determinants of match outcome. Our literature driven approach also identified these variables as being important in women's soccer, with the exception of assists. However, comparisons should be made with caution considering the lower prediction accuracy of the LDA models compared to the DDA and the RA models.

Similar studies have presented models of match outcome in soccer with classification accuracies of 69.5 to 76.9%. This includes models that work as a three-class problem (win, draw, loss) and models that use player-ratings [42]. The relatively high performance of the models (65–99% classification accuracy) in the present work, demonstrates the value of utilising a broad variety of variables when trying to understand the optimal technical characteristics of performance in women's or men's soccer. The present work may also demonstrate the value of training models on relatively large datasets and the use of variable values expressed as relative to the opponent.

This study has some limitations. We used a wide variety of data covering several seasons of two major leagues and two tournaments to increase generalisability to the larger population. However, it is possible that different leagues have different playing styles and that there is a difference in playing style between leagues and tournaments (although we have not assessed this). This could have increased the noise within the data and specific variables important for one league might not be important for another and therefore stay hidden. The initial selection of variables in the rational approach was done by the authors and one could argue that this should have been done by coaches. The feature selection method we used, may retain and remove different variables if used in the future on different data. Furthermore the collinear variables that were excluded from analysis should not be considered unimportant. Finally, our focus on technical performance should not eclipse the importance of other types of factors that affect match outcome (e.g. physical, tactical, etc.).

To conclude we found that the technical actions; scoring first, intentional assists relative to the opponent, the percentage of shots on goal saved by the goalkeeper relative to the opponent, shots on goal relative to the opponent and the percentage of duels that are successful, are the strongest determinants of success in professional women's soccer. These findings will increase the ability for coaches to plan more women-specific match strategies and training plans. We also found that a wider range of variables selected for analysis increases the prediction accuracy of the models. It is important for analyst to collect enough data to create accurate models, but not too much to avoid overfitting.

Recommendations for future research in modelling match outcome based on technical performance in men's soccer are to increase the number and variety of variables used, to include

the use of variables in their relative form, to use a relatively large sample of matches, to apply more accurate modelling techniques and to report the accuracy of the method used. Future research should also make a direct comparison of the modelling of match outcome for both genders within the same study using data mining approaches. Currently, only comparisons between genders on technical performance exists using methods that look at the differences in absolute values of technical variables. The results of the two studies using this type of analysis could indicate a different playing style between the genders. It was found that women players were less accurate in their passes and had higher numbers of lost balls [43, 44], but also have a more attacking style of play. Which is a results of higher number of attacks, interceptions, recoveries and successful challenges and tackles [44]. The more attacking playing style is in line with the results of this study.

Our models included novel variables that provide new insights about optimal technical performance in women's and possibly men's soccer. While a wide range of variables is beneficial, it is also important to avoid redundant data collection in the applied sport science setting [45] and one way to achieve this is to evaluate the importance of variables collected. The feature selection provided an indication of what is and is not important, and the models themselves revealed the most important variables that should be the focus of coaching and feedback. Finally, our findings reinforce the idea that variables that are expressed in their relative form are more important than in their absolute form [23]. Performance analysts and coaches should evaluate the difficulties associated with applying technical performance indicators in their relative form (e.g. not seek to achieve 6 assists in a game, but to seek 2 more assists than their opponent).

## Acknowledgments

We would like to thank Opta Sports for providing the data.

## Author Contributions

**Conceptualization:** Laura M. S. de Jong, Paul B. Gastin, Maia Angelova, Dan B. Dwyer.

**Data curation:** Laura M. S. de Jong.

**Formal analysis:** Laura M. S. de Jong.

**Investigation:** Laura M. S. de Jong, Paul B. Gastin, Lyndell Bruce, Dan B. Dwyer.

**Methodology:** Laura M. S. de Jong, Paul B. Gastin, Maia Angelova, Lyndell Bruce, Dan B. Dwyer.

**Software:** Laura M. S. de Jong.

**Supervision:** Paul B. Gastin, Maia Angelova, Lyndell Bruce.

**Validation:** Laura M. S. de Jong, Dan B. Dwyer.

**Visualization:** Laura M. S. de Jong.

**Writing – original draft:** Laura M. S. de Jong.

**Writing – review & editing:** Laura M. S. de Jong, Paul B. Gastin, Maia Angelova, Lyndell Bruce, Dan B. Dwyer.

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
