## [Decision Letter · Decision Letter 0]

17 Jul 2020

PONE-D-20-13014

Technical determinants of success in professional women’s soccer: A wider range of variables reveals new insights

PLOS ONE

Dear Dr. de Jong,

Thank you for your nice submision to PLOS ONE! After careful consideration, we feel that your manuscript has merit but does not fully meet PLOS ONE’s publication criteria as it currently stands. Therefore, we invite you to submit a revised version of the manuscript that addresses the points raised during the review process.

Both reviewers looked very favourably on your manuscript. They recommended some clarifications and some rewording. Please consider these comments and change the manuscript accordingly.

Both reviewers also indicated that te raw data for this paper was not available. Please make the raw data accessible, since this is one of the lublication criteria of PlosOne.

I'm looking forward to receiving your revisions.

We look forward to receiving your revised manuscript.

Kind regards,

Peter Andreas Federolf

Academic Editor

PLOS ONE

Journal Requirements:

Reviewers' comments:

Reviewer's Responses to Questions

**Comments to the Author**

1. Is the manuscript technically sound, and do the data support the conclusions?

Reviewer #1: Yes

Reviewer #2: Yes

2. Has the statistical analysis been performed appropriately and rigorously? 

Reviewer #1: Yes

Reviewer #2: Yes

3. Have the authors made all data underlying the findings in their manuscript fully available?

Reviewer #1: No

Reviewer #2: No

4. Is the manuscript presented in an intelligible fashion and written in standard English?

Reviewer #1: Yes

Reviewer #2: Yes

5. Review Comments to the Author

Reviewer #1: This is an interesting paper assessing the determinants of match outcome in élite women football. The paper is fairly well structured and conceived.

The methodological approach is sound and nicely explained through informative pictures.

The discussion is relevant and provides interesting insights.

I’ve noticed just a list of minor issues / clarifications that should be fixed before acceptance:

ABSTRACT

rows 16-21: please summarise previous research more succinctly.

Line 21: tautology, “determinants” are “important” by definition.

Line 22: “analytical approached”, there’s something unclear in the sentence. Please define the sample size precisely (not “relatively large sample”).

Line 28: it is unclear what is being predicted.

The last sentence seems a repetition of what stated before.

INTRODUCTION

Line 41: in my opinion, as you’re talking about women football, it’s not necessary to highlight the drawbacks of studies on male players. Stating that such studies on women don’t exist could be a strong enough reason to justify the research.

Line 55: the classification “accuracy”, “power”, etc. A word should follow “classification”.

Lines 60-61 are unnecessary and can be removed.

Line 62: an “s” is missing after “relate”

Line 7: please add a space before “m” (meters)

Line 92: why “understand” has capital letter?

METHODS

I would call the first paragraph “data collection”

An explanation of the variables domain (it’s not feasible to describe every one of the 450 variables) would be strongly advised.

Line 105: so, if I get it right, you should have 266 variables per each team per each game, right? If so, this should be explained more clearly.

Line 105: “variables without values in them” awkward expression.

Line 107: data normality testing should be explained in the data analysis paragraph.

Line 109: there’s no need to explain what R is.

Please remove the “tables” heading.

Line 138: why did you keep 152 variables? Which was the criterion?

Line 159: how did you choose the correlated variable to remove?

Line 188: please remove “was utilised”

DISCUSSION

Line 255: is there anything related to women’s specific psychology in that? I mean, could it be that women when trailing by one goal, could more easily demoralised and struggle to score? Try to clarify / comment this point.

Line 266: just a comment, GK proficiency is probably even more important in women soccer for physical (stature) reasons. Thus, disposing of a tall, athletic and skilled keeper (i.e. one able to save a lot of shots) is a competitive advantage for women (relatively more than in men).

Line 321: that’s true. Why wasn’t it done asking coaches to select variables? Have the Authors themselves first-person experience in women’s football.

Line 328: please remove the comma after “analysis”.

The last sentence could be removed painless.

REFERENCES

Please double check references for consistency (ie. n. 6 but not limited to)

Reviewer #2: General Comments to Author

The authors used three different approaches to identify important technical features/variables in women’s soccer. The authors investigated a high number of variables and a large sample size, which is a major strength of the present manuscript.

Specific comments

Introduction

In the introduction authors focus primary on studies in men’s soccer and the corresponding limitations (lines 39-69). While the comparison of men’s soccer and women’s soccer is of additional interest, the primary focus of the present manuscript should be on studies in women’s soccer and the corresponding limitations (as stated in the aims of the study in lines 92-95). The reviewer suggests shifting the focus on studies in women’s soccer und to reformulate the corresponding sections.

Materials and Methods

Line 105: Please insert the number of variables that were removed (n=10?).

Line 140: At this point it is unclear to the reviewer how the variables relevant to coaches were chosen. As mentioned in the discussion section, variables were not chosen based on coaches feedback. Why?

Line 147: Please be more specific what a “nearly direct relationship” to goal scoring mean. For example, ‘shots on goal’ were excluded while ‘intentional assists’, which result in shots on goal, were excluded.

Lines 150: Please reformulate “more trainable aspects”, because it is not consistent with ‘shots on goal’ (line 148) for example.

Line 159: In the reviewer’s opinion, authors should check these excluded variables. For example, if the features selection process identifies an important variable, which has a high correlation (r>0.9) with one or more of the excluded variables, these excluded variables might also be important.

Line 168: Please be more specific with respect to “reduced model accuracy” because a higher number does not necessarily reduce model accuracy. For example, phase 1 resulted in higher accuracy than phase 2 where lower variables were included.

Line 174: Please specify “MA”, because it was not specified before.

Line 177: Table 2: A reader of the manuscript might benefit from a more detailed description of the listed variables. If possible, the reviewer would suggest providing a more detailed description as supplementary information.

Results

Line 197: Table 3: In Table 3 the listed variables correspond to the GLM model only. This should be indicated somewhere.

Line 203: Please reformulate “slightly higher than random choice”.

Line 206: Table 3 might be more appropriate than Table 2.

Line 208: Please insert almost or another alternative before equally since the models did not perform equally.

Lines 215-227: As noted above it should be noted that the specific variables correspond to the GLM models. Only in lines 235 to 237 specific variables corresponding to the DT approach are presented. Why did the authors decide to exclude a more detailed description of the specific variables of the DT approach in the manuscript?

Lines 231-232: A reader might be interested in the difference between “intentional assists” and “normal assists”. As mentioned above a more detailed description would be beneficial and should be provide as supplementary information, if possible.

Discussion

In the first paragraph should refer to the two objectives (aims) of the study stated at the end of the introduction (i.e., understanding the technical determinants of success in women’ soccer (a) and analytical approaches (b)). Although the comparison to men’s soccer is of interest, it should be discussed in a later paragraph.

Lines 261-264: Because an intentional assists results in a shot on goal, one might also infer that shots on goal preceding an intentional assist are more important than shots on goal in general. Please consider this.

Line 274: Please reformulate “offensive style of play”, because passes refer more to ball possession and (aerial) duels might refer more to the physical condition (i.e., sprinting and jumping abilities). These characteristic might be well addressed in training as well.

Line 290: Please consider that incorporating relative scores might also be important.

Lines 291-303: Please move this paragraph, because recommendations for future research are typically given at the end of the manuscript.

Lines 313-314: Please reformulate the example, because “seeking 2 more assists than their opponent” is not applicable.

Lines 317-319: Did the authors look at the results for a subset of the data to support this hypothesis?

Lines 320-322: See comment above (Line 140).

Lines 315-322: Please consider in the limitations also the following points. First, technical indicators are not the only determinants of success. For example, physical aspect, tactical decisions or psychological factors are also import. Second, please consider also the problem of stability of feature selection (i.e., in the presence of correlations).

6. PLOS authors have the option to publish the peer review history of their article (what does this mean?). If published, this will include your full peer review and any attached files.

Reviewer #1: **Yes: **Matteo Zago

Reviewer #2: No

---

## [Author Response · Author response to Decision Letter 0]

2 Sep 2020

We would like to thank the reviewers for their time and effort, it has certainly strengthened the manuscript.

---

## [Decision Letter · Decision Letter 1]

22 Sep 2020

PONE-D-20-13014R1

Technical determinants of success in professional women’s soccer: A wider range of variables reveals new insights

PLOS ONE

Dear Dr. de Jong,

The reviewers were satisfied with the changes to the manuscript, but one reviewer has some additional recommendations. I invite you to consider these comments.

We look forward to receiving your revised manuscript.

Kind regards,

Peter Andreas Federolf

Academic Editor

PLOS ONE

Reviewers' comments:

Reviewer's Responses to Questions

**Comments to the Author**

1. If the authors have adequately addressed your comments raised in a previous round of review and you feel that this manuscript is now acceptable for publication, you may indicate that here to bypass the “Comments to the Author” section, enter your conflict of interest statement in the “Confidential to Editor” section, and submit your "Accept" recommendation.

Reviewer #2: (No Response)

2. Is the manuscript technically sound, and do the data support the conclusions?

Reviewer #2: Yes

3. Has the statistical analysis been performed appropriately and rigorously? 

Reviewer #2: Yes

4. Have the authors made all data underlying the findings in their manuscript fully available?

Reviewer #2: No

5. Is the manuscript presented in an intelligible fashion and written in standard English?

Reviewer #2: Yes

6. Review Comments to the Author

Reviewer #2: Dear authors,

Thanks very much for addressing my comments (in detail). I only have one remaining remarks:

Introduction:

The authors still focus primary on studies in men's soccer and the corresponding limitations. The reviewer suggests again shifting the focus on studies in women's soccer and to reformulate the corresponding sections.

In particular, the second paragraph (lines 54-60) does not add important information because it focuses on men's soccer and the problem of limited sample size. The problem of limited sample size is already mentioned in the first paragraph (lines 51-53).

In the third paragraph, the authors focus on the technical variables used in previous studies in men’s soccer (i.e., lines 61-64). Why? The present reviewer recommends to focus on previous studies in women’s soccer and the corresponding technical variables.

In the fourth paragraph, the authors focus on statistical models in men's soccer (i.e., lines 74-75). As above, it is recommended to focus on previous studies in women’s soccer and the corresponding statistical models.

In the fifth paragraph, the restricted transferability of the results of studies in men's soccer (to women's soccer) is addressed. Here, further limitations of studies on men's soccer might be listed by the authors, if these are classified as important.

7. PLOS authors have the option to publish the peer review history of their article (what does this mean?). If published, this will include your full peer review and any attached files.

Reviewer #2: No

---

## [Author Response · Author response to Decision Letter 1]

2 Oct 2020

We thank the reviewer for their final comments about the introduction.

---

## [Editor Report · Decision Letter 2]

7 Oct 2020

Technical determinants of success in professional women’s soccer: A wider range of variables reveals new insights

PONE-D-20-13014R2

Dear Dr. de Jong,

We’re pleased to inform you that your manuscript has been judged scientifically suitable for publication and will be formally accepted for publication once it meets all outstanding technical requirements.

Kind regards,

Peter Andreas Federolf

Academic Editor

PLOS ONE

Additional Editor Comments (optional):

Thank you for your submission. I accept your arguments for not changing some of the points that were suggested in the second round of review. With this email I have accepted your manuscript and forwarded it to for production. 

Kind regards,

PF

---

## [Editor Report · Acceptance letter]

13 Oct 2020

PONE-D-20-13014R2 

Technical determinants of success in professional women’s soccer: A wider range of variables reveals new insights 

Dear Dr. de Jong:

I'm pleased to inform you that your manuscript has been deemed suitable for publication in PLOS ONE. Congratulations! Your manuscript is now with our production department. 

Kind regards, 

on behalf of

Dr. Peter Andreas Federolf 

Academic Editor

PLOS ONE